# Development and Applications of CRISPR/Cas9-Based Genome Editing in *Lactobacillus*

**DOI:** 10.3390/ijms232112852

**Published:** 2022-10-25

**Authors:** Yulin Mu, Chengxiao Zhang, Taihua Li, Feng-Jie Jin, Yun-Ju Sung, Hee-Mock Oh, Hyung-Gwan Lee, Long Jin

**Affiliations:** 1College of Biology and the Environment, Nanjing Forestry University, Nanjing 210037, China; 2BioNanotechnology Research Centre, Korea Research Institute of Bioscience & Biotechnology (KRIBB), Daejeon 34141, Korea; 3Cell Factory Research Centre, Korea Research Institute of Bioscience & Biotechnology (KRIBB), Daejeon 34141, Korea

**Keywords:** CRISPR/Cas9, Cas9 protein, *Lactobacillus*, genome editing

## Abstract

*Lactobacillus*, a genus of lactic acid bacteria, plays a crucial function in food production preservation, and probiotics. It is particularly important to develop new *Lactobacillus* strains with superior performance by gene editing. Currently, the identification of its functional genes and the mining of excellent functional genes mainly rely on the traditional gene homologous recombination technology. CRISPR/Cas9-based genome editing is a rapidly developing technology in recent years. It has been widely applied in mammalian cells, plants, yeast, and other eukaryotes, but less in prokaryotes, especially *Lactobacillus*. Compared with the traditional strain improvement methods, CRISPR/Cas9-based genome editing can greatly improve the accuracy of *Lactobacillus* target sites and achieve traceless genome modification. The strains obtained by this technology may even be more efficient than the traditional random mutation methods. This review examines the application and current issues of CRISPR/Cas9-based genome editing in *Lactobacillus*, as well as the development trend of CRISPR/Cas9-based genome editing in *Lactobacillus*. In addition, the fundamental mechanisms of CRISPR/Cas9-based genome editing are also presented and summarized.

## 1. Introduction

*Lactobacillus* is a genus of rod-shaped, non-spore-forming, Gram-positive bacteria that produce large amounts of lactic acid from fermentable carbohydrates [1]. *Lactobacillus* belongs to *Firmicutes*, *Bacilli*, *Lactobacillales*, *Lactobacillaceae*, which is an important genus of the *Lactobacillaceae* [2]. *Lactobacilli* are found in plants and plant products, silage, dairy products, fermented foods, as well as in the oral cavity, vagina, and gut of humans and animals [3]. *Lactobacilli* are not only one of the symbiotic bacteria in the human body, but they are also essential to human health and the food industry.

*Lactobacilli* are generally recognized as safe (GRAS) microorganisms and have a wide range of industrial applications [4,5]. The application of *Lactobacillus* in food fermentation has a long history and it can be used to produce fermented dairy products, meat, and vegetable food, such as yogurt, cheese, sausage, and pickles [6,7,8,9]. *Lactobacilli* utilize fermentable saccharides to produce mild sour lactic acid, and they can also produce other organic acids, such as propionic acid, formic acid, and acetic acid, which constitute the main flavor components of food [10]. While giving the product a sour taste, they can also interact with alcohols, aldehydes, ketones, and other substances produced in the process of lactic acid fermentation to form a variety of new flavoring substances, which constitute the aromatic components in food [11]. *Lactobacillus* can also be used as probiotics, with a variety of prebiotic functions (Table 1). The global probiotic market is anticipated to reach ~92 billion USD in sales by 2026, and the probiotic supplement industry is anticipated to grow at a compound annual growth rate (CAGR) of over 8% between 2021 and 2026 [12]. Therefore, *Lactobacillus*, as a potential candidate for probiotics, has substantial economic significance.

Gene editing is a revolutionary genetic engineering technique that precisely modifies particular target genes in the genomes of organisms [13]. The Clustered Regularly Interspaced Short Palindromic Repeats (CRISPR)/CRISPR-related protein (Cas) system [14] has undergone constant development over the past few years, and is now in its third generation. In comparison to the first-generation Zincfinger Nucleases (ZFNs) and the second-generation Transcription Activator-like Effector Nucleases (TALENs), the CRISPR/Cas9 technology is the most effective and feasible and the least expensive [13,15,16]. The CRISPR/Cas system is an acquired immune system that prokaryotes utilize to resist the invasion of exogenous genetic elements present in bacteriophages or plasmids, and it is a defense mechanism present in most bacteria and all archaea to eliminate alien plastid or phage DNA (Figure 1) [17]. The core of this system consists of two parts: CRISPR sequence and Cas gene [18]. CRISPR is a series of tandem repetitive DNA fragments accidentally found in the genome of *E. coli* [19]. Subsequently, it was found that such repetitive sequences are widely found in the genome of prokaryotes and archaea, and they were named as Clustered Regularly Interspaced Short Palindrome (CRISPR) sequences, and Cas is a related protein with nuclease activity [20,21]. The RNA corresponding to the spacer in CRISPR sequence can be used to identify and cut specific complementary DNA strands [22]. The CRISPR/Cas systems are divided into two classes and six types based on the structure and function of the Cas protein. Class I includes types I, III, and IV, and class II includes types II, V, and VI. The CRISPR/Cas9 system belongs to the type II system, and has developed rapidly as the widely used tool in the precise genome editing [23,24,25].

Currently, CRISPR/Cas9 technology is utilized infrequently on *Lactobacillus*. Utilizing CRISPR/Cas9 for effective gene editing of *Lactobacillus* to strengthen strains with prebiotic properties or to introduce additional prebiotic functionalities as needed would expand the application of *Lactobacillus* in food development, disease treatment, and nutritional wellness. This review will summarize the development and applications of CRISPR/Cas9-based genome editing in *Lactobacillus* over the past few years, and advances and applications of CRISPR/Cas9 technology and its prospects for this genus will be described.

## 2. CRISPR/Cas9 Genome Editing System

### 2.1. Components of CRISPR/Cas9 Genome Editing System

Cas9, crRNA, and tracrRNA are the three components that comprise the natural CRISPR/Cas9 system [33]. Through local base pairing, crRNA and tracrRNA form guide RNA (gRNA), and gRNA binds to Cas9 protein to guide Cas9 protein to specifically recognize and cleave target DNA sequences (Figure 2A) [34]. Researchers fused crRNA and tracrRNA into one RNA, and called it single guide RNA (sgRNA) in order to facilitate experimental design and improve the stability of gRNA (Figure 2B) [35]. The modified CRISPR/Cas9 system has become the preferred tool for researchers for gene editing. For Cas9 protein to successfully recognize the target sequence, two conditions must be met: (1) base pairing between 20 NT at the 5’-end of sgRNA and target DNA; (2) presence of an appropriate PAM sequence at the 3’-end of target DNA. In the CRISPR/Cas9 system, the Cas9 protein contains HNH and RuvC domains that cleaves target DNA to generate DNA double-strand breaks (DSBs) [36,37]. The sgRNA can be engineered to recognize upstream sequences to PAM (NGG); therefore, the CRISPR/Cas9 system has increasingly become an accessible tool.

### 2.2. Mechanism of CRISPR/Cas9 Genome Editing System

Gene editing technology based on CRISPR/Cas9 is mainly divided into two processes (Figure 3), i.e., DNA cleavage and DNA repair [38]. 

DNA cleavage: Cas9 is recruited by guide RNA and binds to guide RNA to form a complex, Cas9 nuclease activity is activated, and the formed complex can start to look for the complementary target DNA sites for recognition and cleavage, and the Cas9 protein specifically cleaves the target DNA sequence to generate double-strand breaks (DSBs) [39,40]. Target recognition and cleavage requires not only complementary pairing of the original 20-nt spacer sequence in the target DNA and the spacer on the guide RNA, but also the existence of a conserved PAM motif in the target attachment [41]. A single mutation in PAM in vitro can lead to loss of Cas9 cleavage activity and enable phages to evade host immune responses [42]. The commonly used SpCas9 (Cas9 nuclease from *Streptococcus pyogenes*) PAM sequence is 5’-NGG-3’, where N can be any of the four DNA bases. 

DNA repair: This mechanism in the cell repairs the DSBs and modifies the DNA sequence during the repair process [43]. DNA repair mechanisms are divided into two categories: homology-directed repair (HDR) and non-homologous end joining (NHEJ) [44]. HDR requires the help of repair templates, which can achieve accurate and controllable editing; NHEJ repair does not rely on repair templates and directly splices two DNA ends [45,46,47]. However, base insertion or deletion (indels) may occur during the splicing process, and accurate editing cannot be achieved. In *Lactobacillus*, the introduced DSBs are usually repaired by HDR, not NHEJ [48].

## 3. Development of CRISPR/Cas9-Based Genome Editing in *Lactobacillus*

### 3.1. Modifications of Cas9 for Genome Editing

Cas9 proteins with nuclease activity are found in the majority of bacteria. The SpCas9 produced from *Streptococcus pyogenes* is the most commonly utilized Cas9 for gene editing [49]. Cas9 protein contains two characteristic nuclease domains (Figure 4A): HNH domain and RuvC domain, inducing DSBs in the target DNA, where the RuvC domain cleaves the non-complementary strand of DNA and the HNH domain cleaves the DNA strand that is complementary to crRNA [50]. The intensity of nuclease activity depends largely on the binding efficiency of Cas9 [51]. Subsequently, some Cas9 mutants emerged to improve gene editing efficiency [52].

Ding et al. demonstrated that SpCas9 could be modified by fusing with chromatin regulatory peptides (CMPs) from high mobility group proteins HMGN1 and HMGB1, histone H1, and the chromatin remodeling complex, and that its activity could be multiplied by several orders of magnitude, particularly on refractory targets [53]. Additionally, the CMPs fusion strategy (CRISPR chrome) is also effective in enhancing the activity of the newly characterized Cas9 homolog of *L. rhamnosus*. At the same time, this CRISPR chrome strategy can be used to improve the established Cas9 nuclease and encourage the discovery of new Cas9 homologues for genome modification.

Cas9 nickase (Cas9n) consists of inactivating mutations in the endonuclease domain [54]. For instance, the Cas9^D10A^ mutation eliminates the endonuclease activity of the RuvC domain while retaining one HNH domain function (Figure 4B); likewise, the Cas9^H840A^ mutation can eliminate the function of the HNH domain while retaining the RuvC function (Figure 4C), which is capable of forming single-strand break (SSB) at the target position [54,55,56]. Goh et al. applied the gene editing method in *L. acidophilus* using the established Cas9^D10A^ nickase variant (Cas9n) [57]. The obtained pLbCas9^N^ system has been further demonstrated in *L. gasseri* and *L. paracasei*. The system (pLbCas9n) has high gene editing efficiency and versatility. It can function flexibly at different chromosomal loci, generate deletions of varying sizes, and construct repair templates for many concurrent mutations. Deletion mutants can be recovered within one week after transformation. Additionally, in some cases, edited cells are present with unedited cells, and subsequent purification can be applied to mixed genotype cultures to recover pure mutant populations.

### 3.2. Guide RNA Expression

The specificity of the CRISPR/Cas9 system for DNA recognition is not determined by proteins, but by 20nt guide RNA [39,58]. By artificially designing crRNA and tracrRNA, sgRNA (single guide RNA), with a guiding effect, can be transformed to guide Cas9 to cleave DNA at the target region, and the rational design of sgRNA can effectively reduce the off-target effects [59,60,61]. At the constant expression level of Cas9 nuclease, the amount of sgRNA transcription was positively correlated with the cleavage efficiency of Cas9 sgRNA complex on DNA. Therefore, selecting strong promoters to enhance the amount of sgRNA transcription can improve gene editing efficiency [62]. Promoter P11 is a synthetic sequence of rRNA promoter from *L. plantarum* WCFS1, which is a strong constitutive promoter. Studies have shown that replacing the strong promoter P11 with the promoter PLP _0537 can enhance the amount of sgRNA transcription, hence improving the editing efficiency of *L. plantarum* WCFS1 [63].

### 3.3. CRISPR/Cas9-Assisted ssDNA Recombineering

Recombineering is a genetic engineering technique based on homologous recombination that can edit the bacterial genome [64]. Double-stranded DNA (dsDNA) and single-stranded DNA (ssDNA) recombineering are included in recombination-mediated genetic engineering [65]. Oh et al. applied CRISPR/Cas9 technology to *Lactobacillus* for the first time, and thereby, developed and optimized CRISPR/Cas9 technology. They combined CRISPR/SpCas9 with ssDNA recombination to knock out three target genes, i.e., *lacL*, *srtA*, and *sdp6*, in *L. reuri* ATCC PTA 6475, and the mutation rate can reach 90%–100% [66].

In most cases, phase recombinase is required for recombineering. However, investigations have demonstrated that phase recombinase is not required [48]. Leenay et al. used CRISPR/Cas9-assisted ssDNA recombineering, exogenous recombinase-free genome editing to simplify CRISPR/Cas9-assisted ssDNA recombineering [67]. This approach enabled the editing of three genes in *L. plantarum* WJL, including the addition of a presequence stop codon in the *ribB* gene associated with vitamin B2 production and the introduction of multiple point mutations in the *ackA* gene encoding acetate kinase. The *lacM* gene involved in *β*-galactosidase production was removed to examine whether this method produces additional mutations in other genes of *L. plantarum* WJL and to determine whether this technique may produce entire gene deletions. This exogenous recombinase-free technique simplifies *L. plantarum* WJL genome editing.

### 3.4. CRISPR/Cas9-Assisted dsDNA Recombineering

Zhou et al. demonstrated CRISPR/Cas9-assisted recombination of double-stranded DNA (dsDNA) in *L. plantarum* WCFS1 designed to knock-out the *nagB* gene which encodes glucosamine-6-phosphate (GlcN-6P) isomerase [68]. Thiophosphate modification was employed to improve dsDNA insertion in order to better optimize the editing procedure. In this way, *L. plantarum* WCFS1 was enhance to produce *N*-acetylglucosamine synthesis (GlcNAc). The efficiency of gene knock-out, gene knock-in, and point mutations is increased to 53.3%, 58.3%, and 62.5%, respectively.

Vento et al. utilized two *E. coli*-*Lactobacilli* shuttle vectors: one containing SpCas9, tracrRNA, and single spacer CRISPR array; the other containing a dsDNA recombinant engineering template [69]. *L. plantarum* was transformed with an *E. coli*-*Lactobacillus* shuttle vector carrying a recombineering template (RT) and homologous arm (HA), and *L. plantarum* was edited using this technique. It is a straightforward and rapid approach of gene editing. Before the correct mutant can be screened, genome editing requires two consecutive transformations of *Lactobacillus*, which can be accomplished in 10 days.

### 3.5. RecE/T-Assisted CRISPR/Cas9 System

The RecE/T recombination system was found in *E. coli* and encoded by *RecE* and *RecT* genes [70]. *RecE* is a 5′-3′ dsDNA exonuclease that cleave exogenous dsDNA to generate a 3′-ended single stranded DNA overhang, and *RecT* is a single-strand annealing protein (SSAP) that binds to the ssDNA overhangs and promotes strand exchange and strand invasion [71,72]. Huang et al. coupled phage derived RecE/T with CRISPR/Cas9, and established a universal toolbox containing recombinant auxiliary plasmids and broad-spectrum host CRISPR/Cas9 editing plasmids [63]. This toolbox can perform effective genome editing in *L. plantarum* WCFS1 and *L. brevis* ATCC367. The RecE/T-assisted CRISPR/Cas9 toolbox achieves single gene deletion with 50–100% efficiency in seven days.

## 4. Applications of CRISPR/Cas9-Based Genome Editing in *Lactobacillus*

As whole genome sequencing technology has advanced, it has been discovered that certain genes in the *Lactobacillus* genome have important functions. These functions include the metabolism of sugars, the production of bacteriocin and extracellular polysaccharides, protein hydrolysis, and phage resistance, etc. [72,73]. Besides, *Lactobacillus* has good resistance to an adverse environments; thus, *Lactobacillus* can be used as a cell factory for biological purification and production of bioactive compounds [74]. In addition, several *Lactobacilli* have demonstrated probiotic properties. CRISPR/Cas9-based genome editing technology is used to develop improved strains with the intent of enhancing the properties of *Lactobacilli*, enhancing the quality and quantity of products, or enhancing and expanding probiotic functions.

Recently, CRISPR/Cas9-based genome editing technology was applied to several *Lactobacilli* (Table 2). *Lactobacillus* can produce two kinds of lactic acid at the same time, l-lactic acid and d-lactic acid. d-lactate dehydrogenase gene (*ldhD*) and l-lactate dehydrogenase gene (*ldhl*) are the key genes that regulate the production of these two types of lactic acid [75]. Tian et al. knocked-out the *ldhD* gene and introduced the *ldhL1* gene through CRISPR/Cas9 genome editing system to transform *L. paracasei* into a high optical purity l-lactic acid producing strain, which could efficiently produce L-lactic acid under 45 °C growth condition [76]. Wang et al. developed single, double, and triple *bsh* knock-out strains using CRISPR/Cas9 genome editing technology in order to determine the significance of different *bsh* genes in bile salt resistance of *L. plantarum* AR113 [77]. The results demonstrated that bsh1 and bsh3 are closely associated with bile salt resistance and will aid in the selection of strains with a high tolerance for bile salts.

To prevent *Lactobacillus* from dying following the formation of DSBs, Goh et al. developed a Cas9 nickase-based genome editing technique that reduces strain mortality from double-strand breaks (DSBs) in *L. acidophilus*, *L. casei*, and *L. gasseri* [57]. To prevent cell death caused by DSBs, Song et al. developed a rapid and precise gene editing method in *L. casei*, utilizing Cas9 nickase that only cleaves a single-strand of DNA, with knock-out and knock-in efficiencies ranging from 25% to 62% [78]. Using this method, the researchers eliminated four non-essential genes and inserted a green fluorescent protein reporter gene. The current gene editing technique for *L. casei* relies on plasmid-based homologous recombination, which requires at least 24 days to produce one gene knock-out. The CRISPR/Cas9^D10A^ system decreased the cycle time to nine days. Li et al. utilized the CRISPR/Cas9^D10A^ system to create auxotrophic *Lactobacillus* W56 (*Lactobacillus ∆Alr W56*) and auxotrophic *L. paracasei* HLJ-27 (*Lactobacillus ∆Alr HLJ-27*) [79]. By inserting the *VP4* gene into the genome using a temperature-sensitive gene editing plasmid, the insertional mutant strains *∆VP4/Alr HLJ-27*, *∆VP4/Alr W56*, and *VP4/W56* were subsequently created. The VP4 protein, which is encoded by the *VP4* gene, is an effective antigen for the development of an anti-PoRV (Porcine rotavirus) vaccine. After oral immunization of mice with the recombinant strains, there were significantly higher levels of serum IgG (immunoglobulin G) and mucosal SIgA (secretory immunoglobulin A), indicating that the insertional mutant strains *∆VP4/Alr HLJ-27*, *∆VP4/Alr W56*, and *VP4/W56* were able to induce the production of mucosal and humoral immune responses against PoRV (Porcine rotavirus).

Zhou et al. [68] discovered that encoding DNA adenine methylase (Dam) in *L. plantarum* WCFS1 improved the efficiency of CRISPR/Cas9-assisted ssDNA recombination by further optimizing the point mutation method. The introduction of Dam increased the efficiency of point mutation for the *L. lantarum* glmS1 gene to 71.4%. Stout et al. identified the function of the CRISPR/Cas9 system in *L. gasseri* JV-V03 and *L. gasseri* NCK1342 and initially noticed the existence of escapees for the off-target effect of Cas9 plasmid transformation [80]. The mechanism of CRISPR-targeted escape was subsequently investigated in further detail. Plasmid interference tests were used to explore the targeting and escape mechanisms of CRISPR/Cas9 system, and the results showed that spacer deletions are the primary escape mechanism in CRISPR/Cas9 system. This research will contribute to a better understanding of the primary causes behind the occasional targeting failures of the CRISPR/Cas9 system.

## 5. CRISPR/Cas9-Based Transcriptional Regulation in *Lactobacillus*

The CRISPR/Cas9 technology can regulate gene expression in addition to being utilized for genome editing. Dead Cas9 (dCas9), nuclease-null Cas9, inhibits the enzymatic activity of RuvC and HNH domains in Cas9 nuclease (Figure 4D) [82]. However, it can still bind to the double strand, hindering the binding of RNA polymerase to the target gene and becoming an important tool for gene silencing [83]. The dCas9 system has been developed as a tool that enabled transcription regulation, epigenetic modifications, DNA looping, and genome imaging [84,85,86]. Based on the dCas9 principle, CRISPR activation (CRISPRa) can be utilized for transcriptional activation, whereas CRISPR interference (CRISPRi) technology has been created to achieve transcriptional suppression [82,87,88]. For bacteria to retain their native cellular metabolism, transcriptional regulation is necessary [89]. The time needed for metabolic engineering is greatly decreased due to the flexible multiplexed gene silencing provided by the CRISPRi method without the need for DSBs induction. Therefore, CRISPRi offers exceptional benefits for regulating important gene expression, examining undiscovered gene functions, and screening target genes at the genome level [89]. Myrbraten et al. developed a two-plasmid CRISPRi system in which dCas9 and gene-specific unidirectional guide RNA (sgRNA) were expressed on different plasmids and could effectively inhibit the expression of any target genes [81]. Using the CRISPRi method to knockdown the phenotype of the genes producing cell wall hydrolase *acm2*, DNA replication start gene *DnaA*, and early cell division protein *ezrA* in *Lactobacillus*, the function of critical cell cycle genes in *L. plantarum* was discovered. This study offers the ideal illustration of how to rapidly screen for critical and non-critical genes using CRISPRi.

## 6. Future Perspectives

As a microorganism closely tied to human life, *Lactobacillus* plays a crucial function in agriculture, industry, and medicine. Therefore, it is essential to develop unique and superior *Lactobacilli* that fulfill human needs. To increase the performance of cell factories in synthetic biology, CRISPR/Cas9-based genome editing technology may perform efficient and selective gene editing in bacteria. However, it is rarely applied to *Lactobacillus*.

### 6.1. Improvement of Immune Regulation Traits

Most *Lactobacilli* have potential immune regulation functions, and CRISPR/Cas9-based genome editing can be used to transform *Lactobacillus* and enhance its immune regulation characteristics. Thomas et al. found out that *L. reuteri*-derived histamine can inhibit the production of human TNF (a pro-inflammatory cytokine) as an immune protein in terms of mediating cellular inflammation [90]. Through the H2 receptor, histamine activates AC (adenylate cyclase) and increases intracellular cAMP (cyclic adenosine monophosphate) levels, which can decrease TNF synthesis at the transcriptional level. Histamine is produced via histidine decarboxylase-catalyzed decarboxylation of histidine. Therefore, the expression of the amino acid decarboxylase gene cluster is essential for histamine synthesis. If CRISPR/Cas9-based genome editing is utilized to increase the expression of this gene, the immune regulation functions of these strains can be boosted in a stable and durable manner.

### 6.2. Increased Resistance to Environmental Stress

*Lactobacilli* are typically fermented in a harsh industrial environment in the food sector. Additionally, after being consumed by humans as probiotics, they must compete with the gut microbiota and the highly dynamic gastrointestinal environment. Using CRISPR/Cas9, the genome of *Lactobacillus* was modified to make it more stable and adaptable to many environmental circumstances.

*Lactobacillus* produces lactic acid and other organic acids through fermentation, and the accumulated acids will affect their metabolic growth and function. Therefore, *Lactobacillus* needs to have acid resistance. The glutamate decarboxylase (GAD) system is an important acid tolerant system of *Lactobacillus* [91]. Gong et al. discovered that the transcriptional regulator GadR has a positive regulatory effect on the system, and that the high expression of the *gadR* gene can improve *L. brevis*’ acid tolerance [92]. Following the construction of an in-frame *glnR* deletion strain of *L. brevis*, it was discovered that the GlnR protein produced by the *glnR* gene can negatively control the glutamate decarboxylase system and that the deletion of *glnR* in *L. brevis* results in increased acid tolerance [93]. Therefore, hyper-expression of the *gadR* gene or knock-out of the *glnR* gene using CRISPR/Cas9-based genome editing may be an effective method to enhance the stress resistance of the strain.

### 6.3. Development of Efficient Mucosal Vaccines

The best way to induce innate and adaptive mucosal immune responses is to immunize the mucosa directly, rather than through the systemic route (parenteral injection). Mucosal vaccines can also induce serum antibodies and systemic cell-mediated responses [94]. Due to its ease of administration and mucosal co-immune system, which allows the induction of immune responses on one mucosal surface and then the movement of immune cells to other distant mucosal sites, mucosal delivery is a particularly attractive vaccination strategy [95]. Therefore, there is a need to develop a vector to protect vaccine antigens from the adverse mucosal environment and deliver them to mucosa-associated lymphoid tissues. *Lactobacillus*, as a safe bacteria that can survive in the human body environment and with the ability to regulate the immune system, can be regarded as a potential vaccine carrier [96].

Trichinellosis is a food-borne parasitic disease that can infect both humans and animals, causing serious economic losses to the animal husbandry and food industry, and also endangering human health. Therefore, the development of a vaccine against *Trichinella spiralis* infection is crucial [97]. Recombinant *L. plantarum* containing TsCPF1 antigen and IL-4 (interleukin-4) was constructed by Xue et al. [98]. Oral immunization in mice significantly stimulated the production of anti-TsCPF1-specific IgG antibody, strong mucosal IgA reaction, and Th1/Th2 mixed immune response, thereby reducing intestinal injury significantly. TsCPF1 is an effective candidate antigen against *Trichinella spiralis* infection, while IL-4 is an excellent adjuvant against *Trichinella spiralis*. In addition, invasive *L. plantarum* carrying TsCPF1 and IL-4 antigens showed the best protective effect and served as a new vaccine against *T. spiralis* infection.

Therefore, through CRISPR/Cas9-based genome editing, the required antigen and adjuvant were cloned into the plasmid as template donors with the Cas9 complex, and properly inserted into the bacteria, which could be applied to treat disease.

## 7. Conclusions and Discussion

As a microorganism closely associated with human life, *Lactobacillus* plays an essential role in agriculture, industry, medicine, and other fields. It is particularly important to develop some new *Lactobacillus* strains with excellent performance. Although there are numerous species of *Lactobacillus*, conventional screening of *Lactobacillus* is a labor-intensive and time-consuming task. In addition, as more and more *Lactobacillus* genomes are sequenced, a large number of unknown genes must be investigated for their function. However, conventional methods of *Lactobacillus* gene manipulation are labor-intensive and inefficient, so there is an urgent need to develop fast and efficient genetic manipulation tools. In recent years, the technique of producing superior strains through CRISPR/Cas9-based genome editing has captured the interest of a huge number of researchers. Compared with earlier gene editing technologies, CRISPR/Cas9-based genome editing technology has less operational difficulty, low editing cost, and higher targeting efficiency. However, there are few reports of gene editing in *Lactobacillus*, mainly because the homologous recombination ability of *Lactobacillus* is limited, and it is difficult to edit multiple targeted genes at the same time. In addition, *Lactobacilli* are extremely susceptible to death after creating DSBs in several locations. We can, therefore, attempt to use Cas9 nickase for gene editing. Insufficient expression of gRNA will also affect subsequent genome editing, and the expression of gRNA can be increased by replacing strong promoters. Additionally, to reduce the off-target effects of the Cas9 system, the properties of Cas9 protein can be changed to make it more accurate when targeting genes or to increase the targeting effectiveness of Cas9 by creating more suitable PAM sequences. Researchers have also created a series of high-fidelity Cas9 variations to increase the editing fidelity or targeting range of CRISPR/Cas9 system [99]. However, additional experimental verification is needed to confirm their applicability to *Lactobacillus*. Using two CRISPR/Cas9-based genome editing approaches on three strains of *L. plantarum*, Leenay et al. discovered that the success rate of editing the same region in different *L. plantarum* strains using the two methods varied [67]. Therefore, other procedures must be created for multiple strains.

In addition to the external CRISPR/Cas9 system, *Lactobacillus* also possesses an endogenous CRISPR/Cas system that can be exploited for targeted gene editing by developing an appropriate gRNA. Hidalgo-Cantabrana et al. successfully altered *L. crispatus* using the endogenous type I-E CRISPR/Cas system, including gene knockout, gene insertion, and various alterations, such as stop codon insertion and single nucleotide replacement [100].

The application of CRISPR/Cas9-based genome editing to *Lactobacillus* has made some progress thus far. Genome editing using CRISPR/Cas9 to improve *Lactobacillus* may be a promising method. However, *Lactobacillus* is extremely diverse, making it challenging to find a universal gene editing tool that is appropriate for all strains. In the future, it will be important to develop more diverse systems for editing the *Lactobacillus* genome.

## Figures and Tables

**Figure 1 ijms-23-12852-f001:**
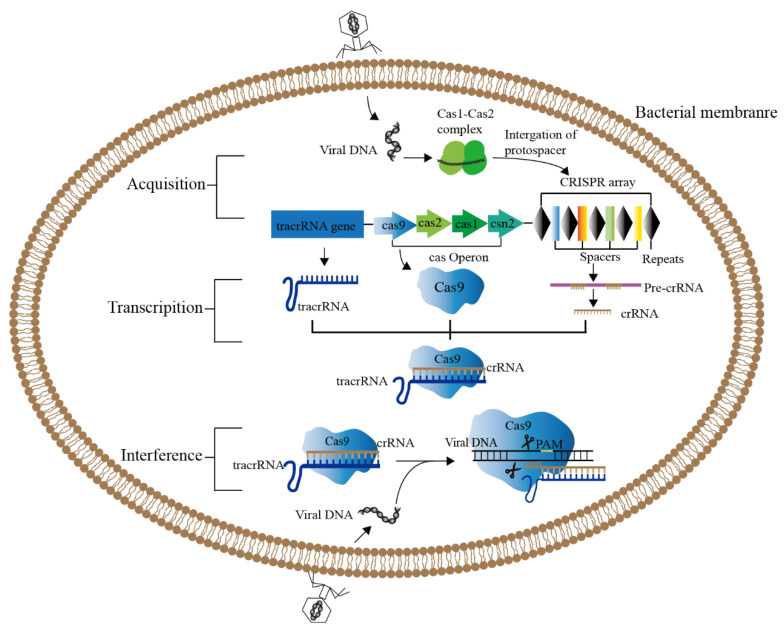
CRISPR/Cas9 system in bacterial adaptive immunity. The functional driving of CRISPR/Cas9 system is divided into three stages. Acquisition: After phage infection, the Cas1-Cas2 protein complex will scan the invading DNA and identify the PAM region, and integrate the new spacer into the CRISPR sequence through the collection mechanism (cas1, cas2, csn2). Transcription: CRISPR sequence is transcribed to produce pre-crRNA under the control of the leading region, and tracrRNA is also transcribed. Pre-crRNA is further processed to form mature crRNA, which forms double-stranded RNA with tracrRNA through base complementary pairing. (3) Interference: The crRNA-tracrRNA duplex binds to Cas9 and guides Cas9 to cleave foreign DNA containing a 20-nt crRNA complementary sequence preceding PAM.

**Figure 2 ijms-23-12852-f002:**
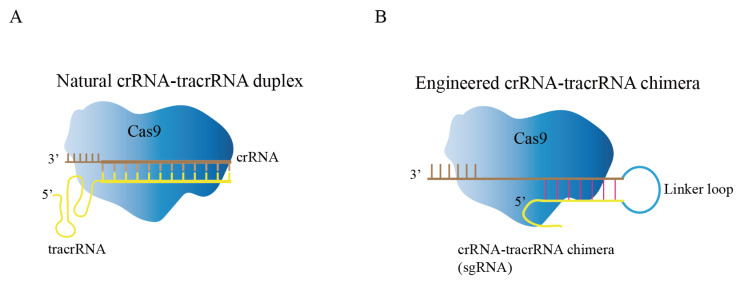
Composition of CRISPR/Cas9 genome editing system. (**A**) Natural CRISPR/Cas9 genome editing system, crRNA and tracrRNA form guide RNA (gRNA) through local base pairing. (**B**) Engineered CRISPR/Cas9 genome editing system; the crRNA-tracrRNA chimera was modified to form sgRNA by creating a connecting loop.

**Figure 3 ijms-23-12852-f003:**
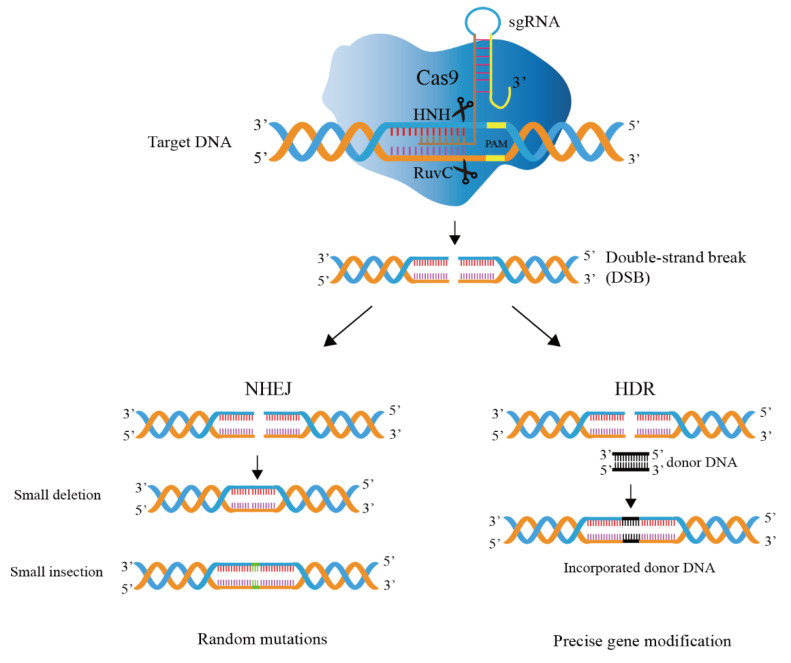
Mechanism of CRISPR/Cas9 genome editing system. The guide RNA directs a Cas9 endonuclease to cleavage target DNA. The DSB (double-strand break) generated by Cas9 nuclease domains is repaired by host-mediated DNA repair mechanisms, i.e., homology-directed repair (HDR) and non-homologous end joining (NHEJ).

**Figure 4 ijms-23-12852-f004:**
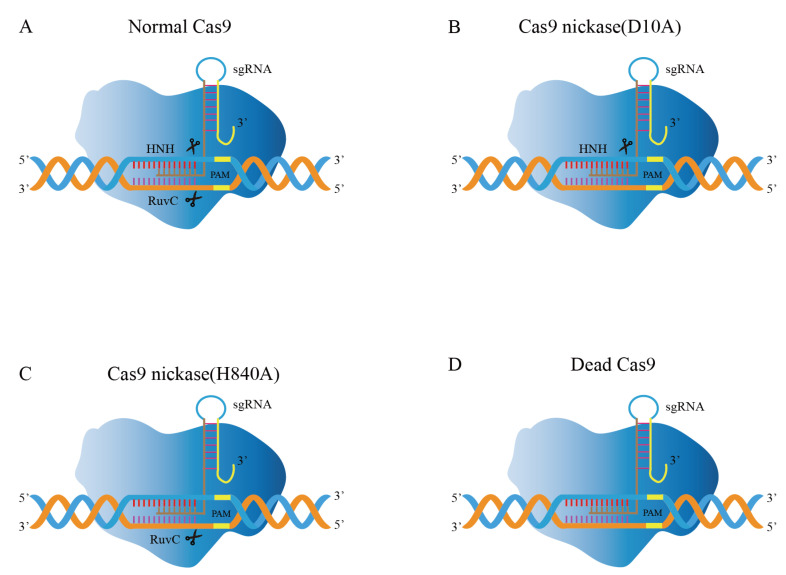
Modifications of Cas9 in genome editing. (**A**) The Cas9 nuclease cleaves the strands of the target DNA via HNH and RuvC domains. (**B**) Cas9n, containing the inactivating mutation D10A, inhibits the activity of the RuvC domain. (**C**) Cas9n, containing the inactivating mutation H840A, inhibits the activity of the RuvC domain. (**D**) Dcas9, inactivating the mutation of HNH and RuvC domains.

**Table 1 ijms-23-12852-t001:** Probiotic function of *Lactobacillus*.

*Lactobacillus* Species	Function	Reference
*Lactobacillus rhamnosus*	Inhibiting cell inflammation and apoptosis	[26]
*L. fermentum*	Antifatigue and antioxidation	[27]
*L. paracasei*	Antitumor	[28]
*L. johnsonii*	Modulating intestinal environment and improving gut development	[29]
*L. pentosus*	Inhibit intestinal pathogenic bacteria	[30]
*L. casei*	Increasing antioxidative capacity	[31]
*L. helveticus*	Cholesterol-lowering activity	[32]

**Table 2 ijms-23-12852-t002:** The development and application of the CRISPR/Cas9-based genome editing in several *Lactobacilli*.

Species	Tools	Plasmids	DNA Repair System	Gene Editing Type	Efficiency	References
*Lactobacillus plantarum*	CRISPR/Cas9 system	pLdbsh1, pLdbsh2, pLdbsh3, pLdbsh4	HDR	gene knock-out	Success	[77]
*L. paracasei*	CRISPR/Cas9 system	pNcas-ΔldhL1, pNcas-ΔldhD-ldhL1	HDR	gene knock-out, gene knock-in	Success	[76]
*L. casei*	CRISPR/Cas9^D10A^ system	pLCNICK	HDR	gene deletion and insertion	25–62%	[78]
*L. acidophilus*	CRISPR/Cas9^D10A^ system	pLbCas9N (pTRK1204 (rafE)), pTRK1205 (lacS), pTRK1254 (ltaS), pTRK1255 (mCherry)	HDR	gene knock-out	100%	[57]
*L. gasseri*	CRISPR/Cas9^D10A^ system	pLbCas9N (pTRK1256 (2crr)	HDR	gene knock-out	100%	[57]
*L. paracasei*	CRISPR/Cas9^D10A^ system	pLbCas9N (pTRK1257 (glgA))	HDR	gene knock-out	100%	[57]
*L. paracasei*	CRISPR/Cas9^D10A^ system	pLCNICK	HDR	gene knock-out	Success	[79]
*L. plantarum*	CRISPRi	pSIP-SH-dCas9, pSgRNA	HDR	gene knockdown	Success	[81]
*L. plantarum*	RecE/T-assisted CRISPR/Cas9 system	pHSP02, pHSP04	HDR	gene deletion, gene replacement/insertion	80–100%	[63]
*L. brevis*	RecE/T-assisted CRISPR/Cas9 system	pHSB01, pHSB04, pHSB05	HDR	gene deletion, gene knock-out	75–100%	[63]
*L. reuteri*	CRISPR/Cas9-assisted ssDNA recombineering	pVPL3004, pVPL3017	HDR	gene mutation, codon saturation mutagenesis	90–100%	[66]
*L. plantarum*	CRISPR/Cas9-assisted ssDNA recombineering	pCas9_RSR	HDR	point mutations	100%	[67]
*L. plantarum*	CRISPR/Cas9-assisted ssDNA recombineering	pCas9_RSR	HDR	silent mutation	Success	[67]
*L. plantarum*	CRISPR/Cas9-assisted ssDNA recombineering	pCas9_RSR	HDR	a complete gene deletion	Success	[67]
*L. plantarum*	CRISPR/Cas9-assisted dsDNA recombineering	pSIP-C9 (nagB), pSIP-C9 (lox), p411-RecT, p411-RecT-Dam	HDR	gene knock-out, insertion, and point mutation	53.3%, 58.3%, 62.5%	[68]
*L. plantarum*	CRISPR/Cas9-assisted dsDNA recombineering	PCB578, pCB591	HDR	gene mutation	Success	[69]

## Data Availability

Not applicable.

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
