# Peer review of "Development and Applications of CRISPR/Cas9-Based Genome Editing in Lactobacillus"

_ijms, 2022, doi:10.3390/ijms232112852_

Round 1

Reviewer 1 Report

More efficient genome editing tools should facilitate both basic research and engineering of lactobacillus. This manuscript, as its title indicated, was to summarize recent progress on CRISPR-Cas based genome editing of lactobacillus. This topic has been extensively reviewed in Roberts et al 2020 and Song et al 2020, as well as the part of Li et al 2021. However, I could not find anything new regarding CRISPR-Cas tools in lactobacillus in the manuscript. Instead, most of the manuscript and all four figures addressed the principle of CRISPR-Cas which should not be the main points according to the title. The authors talked too much in the "future perspective" part about the targets of genome modification for lactobacillus which also did not match the title well. The authors should address the bottlenecks and workable solutions in CRISPR-Cas based genome editing approach in lactobacillus instead of what genetically modified lactobacillus can do.

1. Please provide the evidence for the sentence "The strands observed by this technology may even be safer than the traditional random mutation methods" shown in the abstract.

2. In the part of Cas9 nickase, please verify that it carried the mutation of H840A, not H847A.

3. In part 3.4, please discuss editing efficiency.

4. Please unify the “SpCas9”, as SPCas9 and spCas9 were appeared in this manuscript.

Roberts, A., & Barrangou, R. (2020). Applications of CRISPR-Cas systems in lactic acid bacteria. FEMS Microbiol Rev, 44(5), 523-537.

Song, X., Zhang, X. Y., Xiong, Z. Q., Liu, X. X., Xia, Y. J., Wang, S. J., & Ai, L. Z. (2020). CRISPR-Cas-mediated gene editing in lactic acid bacteria. Mol Biol Rep, 47(10), 8133-8144.

Li, Q., Zhang, J., Yang, J., Jiang, Y., & Yang, S. (2021). Recent progress on n-butanol production by lactic acid bacteria. World Journal of Microbiology and Biotechnology, 37(12), 205.

Author Response

ijms-1953293

Development and Applications of CRISPR/Cas9-Based Genome Editing in Lactobacillus

Reviewer 1

More efficient genome editing tools should facilitate both basic research and engineering of lactobacillus. This manuscript, as its title indicated, was to summarize recent progress on CRISPR-Cas based genome editing of lactobacillus. This topic has been extensively reviewed in Roberts et al 2020 and Song et al 2020, as well as the part of Li et al 2021. However, I could not find anything new regarding CRISPR-Cas tools in lactobacillus in the manuscript. Instead, most of the manuscript and all four figures addressed the principle of CRISPR-Cas which should not be the main points according to the title. The authors talked too much in the "future perspective" part about the targets of genome modification for lactobacillus which also did not match the title well. The authors should address the bottlenecks and workable solutions in CRISPR-Cas based genome editing approach in lactobacillus instead of what genetically modified lactobacillus can do.

Authors’ response: We appreciate your detailed and expert review and will carefully consider it. Your recommendations have helped to increase the uniformity and accuracy. We have taken all necessary steps to improve the manuscript and have made all recommended changes. Regarding the bottlenecks and practical solutions in CRISPR-Cas9 technology, we have discussed and included this information in the main text (L378-401, L404-408).

  1. Please provide the evidence for the sentence "The strands observed by this technology may even be safer than the traditional random mutation methods" shown in the abstract.

Authors’ response: The safety of CRISPR-Cas-based gene editing technology is debatable, but some researchers assert that it is more effective than conventional methods. Therefore, we have modified the original description to remove the safety concern. (L21-22).

  1. In the part of Cas9 nickase, please verify that it carried the mutation of H840A, not H847A.

Authors’ response: Thanks for your detail correction, it was changed as H840A (L158, 173, and Figure 4C).

  1. In part 3.4, please discuss editing efficiency.

Authors’ response: We would like to thank you for your advice, and we have provided additional explanation regarding the effectiveness of editing. (L213-214, 219-221).

  1. Please unify the “SpCas9”, as SPCas9 and spCas9 were appeared in this manuscript.

Authors’ response: Thanks for your kind corrections, all were modified as “SpCas9” (L124, 140, 148, 193, 215).

Roberts, A., & Barrangou, R. (2020). Applications of CRISPR-Cas systems in lactic acid bacteria. FEMS Microbiol Rev, 44(5), 523-537.

Song, X., Zhang, X. Y., Xiong, Z. Q., Liu, X. X., Xia, Y. J., Wang, S. J., & Ai, L. Z. (2020). CRISPR-Cas-mediated gene editing in lactic acid bacteria. Mol Biol Rep, 47(10), 8133-8144.

Li, Q., Zhang, J., Yang, J., Jiang, Y., & Yang, S. (2021). Recent progress on n-butanol production by lactic acid bacteria. World Journal of Microbiology and Biotechnology, 37(12), 205.

Authors’ response:

We agree with reviewer-1’s opinion that this topic had been extensively studied. As a review paper, though lacking in novelty, we put the significance of this paper on Lactobacillus. Lactic acid bacteria (LAB) comprise the Aerococcaceae, Carnobacteriaceae, Enterococcaceae, Lactobacillaceae, Leuconostocaceae, and Streptococcaceae families. In addition to being one of the symbiotic bacteria in the human body, Lactobacillus is also crucial to human health and the food business.

We would like to express our gratitude for the insightful suggestions that you have provided regarding our research paper. These remarks will be included into an effort to enhance the overall quality of the work. We tried our best to improve the manuscript and made all changes that reviewers mentioned in the manuscript (marked blue). These changes will not influence the comment and framework of the paper. We appreciate for your detail and kind work earnestly, and hope that the corrections made here will meet with approval. Once again, thank you very much for your comments and suggestions.

Yours sincerely

Prof. Long Jin

Nanjing Forestry University

Reviewer 2 Report

This manuscript reviewed the CRISPR/Cas9-based gene editing technology in Lactobacillus. Utilizing CRISPR/Cas9 for effective gene editing of Lactobacillus to strengthen strains with prebiotic properties or to introduce additional prebiotic functionalities as needed would expand the application of Lactobacillus in food development, disease treatment, and nutritional wellness. This review introduced the development and applications of CRISPR/Cas9- based genome editing in Lactobacillus over the past few years, looks forward to the development trend of CRISPR/Cas9 technology in the application of Lactobacillus in the future. I think that the subject of this manuscript is fair enough, and the authors seem to cover the most of important previous studies on the topics. However, I raised some concerns regarding the content of this manuscript. The specific comments are below.

1.      The writing of the manuscript need to improve, both in terms of language use but also in terms of logic, flow and scientific accuracy.

2.      Technically speaking, dCas9 mediated CRISPR interference (CRISPRi) technology is not genome editing, it has been adapted by targeting nuclease deficient CRISPR proteins to the transcription regulation region for gene regulation. When coupled with gRNAs libraries, CRISPR-mediated systems can prove to be a valuable tool for genome-wide studies of gene regulation. The author should distinguish this fact. Thus this part should be taken out as a single section.

3.      Part 4, the applications examples of CRISPR/Cas9-based genome editing in Lactobacillus here are not sufficient. Some examples in Part 3 can also be listed and summarized as the applications examples.

Author Response

ijms-1953293

Development and Applications of CRISPR/Cas9-Based Genome Editing in Lactobacillus

Reviewer 2

This manuscript reviewed the CRISPR/Cas9-based gene editing technology in Lactobacillus. Utilizing CRISPR/Cas9 for effective gene editing of Lactobacillus to strengthen strains with prebiotic properties or to introduce additional prebiotic functionalities as needed would expand the application of Lactobacillus in food development, disease treatment, and nutritional wellness. This review introduced the development and applications of CRISPR/Cas9- based genome editing in Lactobacillus over the past few years, looks forward to the development trend of CRISPR/Cas9 technology in the application of Lactobacillus in the future. I think that the subject of this manuscript is fair enough, and the authors seem to cover the most of important previous studies on the topics. However, I raised some concerns regarding the content of this manuscript. The specific comments are below.

  1. The writing of the manuscript need to improve, both in terms of language use but also in terms of logic, flow and scientific accuracy.

Authors’ response: We authors are very grateful for your feedback and will take it into careful consideration. As a result of your suggestions, the consistency and precision have been improved. We have done everything to enhance the manuscript, and we have implemented all of the alterations that were indicated.

  1. Technically speaking, dCas9 mediated CRISPR interference (CRISPRi) technology is not genome editing, it has been adapted by targeting nuclease deficient CRISPR proteins to the transcription regulation region for gene regulation. When coupled with gRNAs libraries, CRISPR-mediated systems can prove to be a valuable tool for genome-wide studies of gene regulation. The author should distinguish this fact. Thus this part should be taken out as a single section.

Authors’ response: We thanks a lot for your insightful recommendation, we have set it as a single part following your advice according to you and another reviewer’s suggestion (Part 5).

  1. Part 4, the applications examples of CRISPR/Cas9-based genome editing in Lactobacillus here are not sufficient. Some examples in Part 3 can also be listed and summarized as the applications examples.

Authors’ response: We appreciate your comments for this section, and the examples of applications were updated according to your suggestion (L255-287).

We appreciate your helpful feedback on our study paper and thank for your positive response on the article. Your comments will be used to enhance the quality of the publication. In response to your kind suggestions, we have gone over the entire manuscript and made the necessary modifications. Please notice that modifications are highlighted in blue across the whole manuscript.

Yours sincerely

Prof. Long Jin

Nanjing Forestry University

Reviewer 3 Report

In this review manuscript, Mu et al describes the CRISPR-based tools for Lactobacilli. I like the concept, presented ideas, organization, figures, and text content. Overall, manuscript is well written but often suffers from minor grammar flaws and inconsistencies. I think that the strength of this manuscript can be further invigorated by addressing following comments.

General comment: it would have been nice to have a line # to submitted version, thus enabling reviewers to refer to the corrections/revisions without having to copy entire statements.

1.     In the abstract line “develop some new lactobacillus” remove “some”

2.     Line in the abstract should read Compared with the traditional strain improvement methods,…

3.     Replace “gram-positive” with “Gram-positive”. 

4.     91,1 billion US dollars should be replaced with “~$92 billion”.

5.     Define CAGR before using this acronym.

6.     Remove “simplest” from line saying…most effective, simplest, and least expensive instead say “feasible”.

7.     All the Figure fonts sizes are too tiny, please increase the font to a legible size (akin to A/B/C etc., panel title size).

8.     Revise “…has the fastest development speed and is widely used as a gene editing technology” to “has developed rapidly as the widely used tool in the precise genome editing”.

9.     Revise “This review will introduce the development” to “This review will summarize the development”

10.  Revise “looks forward to the development trend of CRISPR/Cas 9 technology in the application of Lactobacillus in the future.” to “and advances and applications of CRISPR/Cas9 technology and its prospects for this genus will be described”

11.  Section heeding for “2.1. Composition of CRISPR/Cas9 Genome Editing System” word “composition” sounds odd perhaps say “components”

12.  Please clarify the role of HNH and RuvC individually; this will eventually help readers navigate section 3.

13.  Line: “The sgRNA can be designed to recognize sequences before PAM sequences, so the CRISPR/Cas9 system has become an easy to manage genome editing tool” should be revised to “The sgRNA can be engineered to recognize upstream sequences to PAM (NGG), therefore CRISPR/Cas9 system has increasingly become an accessible tool” Also, word “easy” is quite subjective terms in this context.

14.  Text “in-vitro” should replace current in vitro.

15.  In section 2.2 – “SpCas9” should be defined before writing this acronym.

16.  Author should specify explicitly, if Lactobacilli has any NHEJ or not in section 2.2 with appropriate citation since current references #45-47 are not specific for Lactobacilli.

17.  Please check how SpCas9 is written in the statement and thought out the manuscript “nevertheless,

SPCas9 produced from Streptococcus pyogenes is the most commonly utilized Cas9 for gene editing”

18.  In the section 3.1 about CRISPRi perhaps authors should comment on the use of this tool being widespread for studying genes/processes that are essential (and thus knockout is either impossible or difficult to obtain normally).

19.  Authors should specify if derivative “strong promoter” is constitutive or inducible in this section. “Promoter P11 is a synthetic sequence of rRNA promoter from L. plantarum WCFS1, which is a strong promoter.”

20.  Was stop codon removed and added at the same position or was it at a different position? “This approach enabled the editing of three genes in L. plantarum WJL, including the removal of a premature stop codon in the riboflavin ribB gene. A premature stop codon was added into the ribB gene, which is related with vitamin B2 production, and numerous point mutations were introduced into the ackA gene, which encodes acetic acid kinase.”

21.  For all XYZ et al [citation] sentences, make sure “XYZ et al…. sentence [citation] period punctuation format is followed. For example: Zhou et al demonstrated CRISPR/Cas9-assisted recombination of double stranded DNA (dsDNA) in L. plantarum WCFS1 designed to knock-out the nagB gene which encodes glucosamine-6-phosphate (GlcN-6P) isomerase [75]. Same for Oh et al [72], Leenay et al [74], Vento et al [76], Huang et al. [69], Tian et al [83], Wang et al [84], Thomas et at [86], Gong et al [88], Leenay et al [74] , Hidalgo-Cantabrana et al [95] etc.

22.  There should be a space between protein and (SSAP) “single-strand annealing protein(SSAP)”

23.  “RECE/T-assisted CRISPR/Cas9” should read “RecE/T-assisted CRISPR/Cas9”

24.  In future perspective section, “CRISPR/Cas9-based Genome Editing-based gene editing technology” check capitalization of “genome editing-based”

25.  Replace “editing on bacteria” to “editing in bacteria”

26.  Replace “Improve Immune Regulation Traits” should read “Improvement of Immune Regulation Traits”

27.  Replace “Improve the Resistance to Environmental Stress” to “Increased Resistance to Environmental Stress”

28.  Consider revising this “they must compete with the gastrointestinal environment” to “they must compete with the gut microbiota and the highly dynamic gastrointestinal environment”.

29.  “as a kind of safe bacteria that” remove “kind of” from the statement.

Author Response

ijms-1953293

Development and Applications of CRISPR/Cas9-Based Genome Editing in Lactobacillus

Reviewer 3

In this review manuscript, Mu et al describes the CRISPR-based tools for Lactobacilli. I like the concept, presented ideas, organization, figures, and text content. Overall, manuscript is well written but often suffers from minor grammar flaws and inconsistencies. I think that the strength of this manuscript can be further invigorated by addressing following comments.

General comment: it would have been nice to have a line # to submitted version, thus enabling reviewers to refer to the corrections/revisions without having to copy entire statements.

Authors’ response: Thanks for your kind remind and we have added line numbers through whole manuscript.

  1. In the abstract line “develop some new lactobacillus” remove “some”

Authors’ response: We’ve removed “some” from the sentence (L14).

  1. Line in the abstract should read Compared with the traditional strain improvement methods,…

Authors’ response: corrected (L19).

  1. Replace “gram-positive” with “Gram-positive”.

Authors’ response: corrected (L29).

  1. 91, 1 billion US dollars should be replaced with “~$92 billion”.

Authors’ response: Thanks for your kind correction, the number was replaced (L47).

  1. Define CAGR before using this acronym.

Authors’ response: Full name of CAGR was given in the text (L48).

  1. Remove “simplest” from line saying…most effective, simplest, and least expensive instead say “feasible”.

Authors’ response: replaced with “feasible” (L57).

  1. All the Figure fonts sizes are too tiny, please increase the font to a legible size (akin to A/B/C etc., panel title size).

Authors’ response: The font size of figure and table legends was modified to match that of the main text.

  1. Revise “…has the fastest development speed and is widely used as a gene editing technology” to “has developed rapidly as the widely used tool in the precise genome editing”.

Authors’ response: The sentence was modified as your comments (L70-71).

  1. Revise “This review will introduce the development” to “This review will summarize the development”

Authors’ response: replaced (L76).

  1. Revise “looks forward to the development trend of CRISPR/Cas 9 technology in the application of Lactobacillus in the future.” to “and advances and applications of CRISPR/Cas9 technology and its prospects for this genus will be described”

Authors’ response: Thank you very much for your recommendation; the sentence has been updated (L77-78).

  1. Section heeding for “2.1. Composition of CRISPR/Cas9 Genome Editing System” word “composition” sounds odd perhaps say “components”

Authors’ response: Thanks for your kind comment, and the subtitle was modified (L93).

  1. Please clarify the role of HNH and RuvC individually; this will eventually help readers navigate section 3.

Authors’ response: We’ve made it clear in the text according to your suggestion (L143-145).

  1. Line: “The sgRNA can be designed to recognize sequences before PAM sequences, so the CRISPR/Cas9 system has become an easy to manage genome editing tool” should be revised to “The sgRNA can be engineered to recognize upstream sequences to PAM (NGG), therefore CRISPR/Cas9 system has increasingly become an accessible tool” Also, word “easy” is quite subjective terms in this context.

Authors’ response: Thanks for your good recommendation, the sentence was replaced (L105-107).

  1. Text “in-vitro” should replace current in vitro.

Authors’ response: corrected (L122).

  1. In section 2.2 – “SpCas9” should be defined before writing this acronym.

Authors’ response: The definition of “SpCas9” was given in the manuscript (L124).

  1. Author should specify explicitly, if Lactobacilli has any NHEJ or not in section 2.2 with appropriate citation since current references #45-47 are not specific for Lactobacilli.

Authors’ response: Thanks for your detail and insightful assessment; we have clarified it in the text. (L131-132).

  1. Please check how SpCas9 is written in the statement and thought out the manuscript “nevertheless, SPCas9 produced from Streptococcus pyogenes is the most commonly utilized Cas9 for gene editing”

Authors’ response: Thanks for your kind corrections, all were modified as “SpCas9” (L124, 140, 148, 193, 215).

  1. In the section 3.1 about CRISPRi perhaps authors should comment on the use of this tool being widespread for studying genes/processes that are essential (and thus knockout is either impossible or difficult to obtain normally).

We have implemented your suggestion, along with that of one of the other reviewers, and made it a individual part (Part 5). We have reorganized Part 5 taking into account the recommendation that you made.

  1. Authors should specify if derivative “strong promoter” is constitutive or inducible in this section. “Promoter P11 is a synthetic sequence of rRNA promoter from L. plantarum WCFS1, which is a strong promoter.”

Authors’ response: We’ve clarified it in the text (L184).

  1. Was stop codon removed and added at the same position or was it at a different position? “This approach enabled the editing of three genes in L. plantarum WJL, including the removal of a premature stop codon in the riboflavin ribB gene. A premature stop codon was added into the ribB gene, which is related with vitamin B2 production, and numerous point mutations were introduced into the ackA gene, which encodes acetic acid kinase.”

Authors’ response: The original description appears to be inaccurate. We have rephrased the text in the hopes that it will be easier for readers to comprehend. (L199-202)

  1. For all XYZ et al [citation] sentences, make sure “XYZ et al…. sentence [citation] period punctuation format is followed. For example: Zhou et al demonstrated CRISPR/Cas9-assisted recombination of double stranded DNA (dsDNA) in L. plantarum WCFS1 designed to knock-out the nagB gene which encodes glucosamine-6-phosphate (GlcN-6P) isomerase [75]. Same for Oh et al [72], Leenay et al [74], Vento et al [76], Huang et al. [69], Tian et al [83], Wang et al [84], Thomas et at [86], Gong et al [88], Leenay et al [74] , Hidalgo-Cantabrana et al [95] etc.

Authors’ response: Thank you for your detailed assessment; all of the writings have been revised based on your recommendations (L151, 162, 196, 199, 210, 217, 229, 250, 252, 307,323, 342,402, 410).

  1. There should be a space between protein and (SSAP) “single-strand annealing protein (SSAP)”

Authors’ response: The space problem was figured out (L226).

  1. “RECE/T-assisted CRISPR/Cas9” should read “RecE/T-assisted CRISPR/Cas9”.

Authors’ response: corrected (L230).

  1. In future perspective section, “CRISPR/Cas9-based Genome Editing-based gene editing technology” check capitalization of “genome editing-based”

Authors’ response: Thanks a lot for your detail review, the sentence was modified (316).

  1. Replace “editing on bacteria” to “editing in bacteria”

Authors’ response: replaced (L317).

  1. Replace “Improve Immune Regulation Traits” should read “Improvement of Immune Regulation Traits”

Authors’ response: replaced (L318).

  1. Replace “Improve the Resistance to Environmental Stress” to “Increased Resistance to Environmental Stress”

Authors’ response: replaced (L331).

  1. Consider revising this “they must compete with the gastrointestinal environment” to “they must compete with the gut microbiota and the highly dynamic gastrointestinal environment”.

Authors’ response: Thanks for your suggestion, and sentence was modified (L333-334).

  1. “as a kind of safe bacteria that” remove “kind of” from the statement.

Authors’ response: modified (L358).

We appreciate your positive feedback on the submitted article. The manuscript was revised in accordance with the reviewer's recommendations. These modifications will not affect the paper's discussion or structure. We sincerely appreciate your attention to detail and kindness, and we hope that the corrections made here will be accepted. Thank you again for your thoughtful comments and suggestions.

Yours sincerely

Prof. Long Jin

Nanjing Forestry University

Round 2

Reviewer 1 Report

The author has revised the manuscript carefully, but the modifications to the Figures are not ideal, as all four figures are still the principle of CRISPR-Cas which should not be the main points according to the title.